# Natural infection of parvovirus in wild fishing cats (*Prionailurus viverrinus*) reveals extant viral localization in kidneys

**Chutchai Piewbang**[1,2], **Sabrina Wahyu Wardhani**[2,3], **Jira Chanseanroj**[4], **Jakarwan Yostawonkul**[3,5], **Suwimon Boonrungsiman**[5], **Nattika Saengkrit**[5], **Piyaporn Kongmakee**[6], **Wijit Banlunara**[1], **Yong Poovorawan**[4], **Tanit Kasantikul**[7], **Somporn Techangamsuwan**[1,2]*

1 Department of Pathology, Faculty of Veterinary Science, Chulalongkorn University, Bangkok, Thailand, 2 Animal Virome and Diagnostic Development Research Group, Faculty of Veterinary Science, Chulalongkorn University, Bangkok, Thailand, 3 The International Graduate Course of Veterinary Science and Technology (VST), Faculty of Veterinary Science, Chulalongkorn University, Bangkok, Thailand, 4 Center of Excellence in Clinical Virology, Faculty of Medicine, Chulalongkorn University, Bangkok, Thailand, 5 National Nanotechnology Center (NANOTEC), National Science and Technology Development Agency (NSTDA), Pathumthani, Thailand, 6 The Zoological Park Organization Under The Royal Patronage of H.M. The King, Bangkok, Thailand, 7 Clemson Veterinary Diagnostic Center, Clemson University, Columbia, South Carolina, United States of America

* somporn62@hotmail.com

**Data Availability Statement:** Obtained CPPV-1 sequences of fishing cat nos 1 & 3 have been deposited in NCBI GenBank under accession

## Abstract

Carnivore protoparvovirus-1 (CPPV-1), a viral species containing feline panleukopenia virus (FPV) and canine parvovirus (CPV) variants, are widely spread among domestic and wild carnivores causing systemic fatal diseases. Wild fishing cats (*Prionailurus viverrinus*), a globally vulnerable species, have been found dead. Postmortem examination of the carcasses revealed lesions in intestine, spleen and kidney. CPPV-1 antigen identification in these tissues, using polymerase chain reaction (PCR) and immunohistochemistry (IHC), supported the infection by the virus. PCR- and IHC-positivity in kidney tissues revealed atypical localization of the virus while *in situ* hybridization (ISH) and transmission electron microscopy (TEM) with the pop-off technique confirmed the first description of viral localization in kidneys. Complete genome characterization and deduced amino acid analysis of the obtained CPPV-1 from the fishing cats revealed FPV as a causative agent. The detected FPV sequences showed amino acid mutations at I566M and M569R in the capsid protein. Phylogenetic and evolutionary analyses of complete coding genome sequences revealed that the fishing cat CPPV-1 genomes are genetically clustered to the FPV genomes isolated from domestic cats in Thailand. Since the 1970s, these genomes have also been shown to share a genetic evolution with Chinese FPV strains. This study is the first evidence of CPPV-1 infection in fishing cats and it is the first to show its localization in the kidneys. These findings support the multi-host range of this parvovirus and suggest fatal CPPV-1 infections may result in other vulnerable wild carnivores.

numbers MW145540-MW145541, respectively. All other relevant data are within the manuscript and its Supporting Information files.

**Funding:** C.P. was supported by the Ratchadapisek Somphot Fund for Postdoctoral Fellowship, Chulalongkorn University. S.W. is financially afforded by scholarship program for ASEAN countries, Chulalongkorn University. This research was funded by The Thailand Research Fund (RSA6180034), Grant for Joint Funding of External Research Project, Ratchadaphisek Somphot Endowment Fund and Veterinary Science Research Fund (RES_61_364_31_037), Chulalongkorn University, and Veterinary Pathogen Bank, Faculty of Veterinary Science, Chulalongkorn University.

**Competing interests:** The authors have declared that no competing interests exist.

## Introduction

The genus *Prionailurus*, family *Felidae*, is categorized as a group of spotted, small-sized, wild cats that are native to Asia [1]. The *Prionailurus* genus comprises four formally recognized wild cat species including the leopard cat (*P. bengalensis*), the flat-head cat (*P. planiceps*), the rusty-spotted cat (*P. rubiginosus*) and the fishing cat (*P. viverrinus*) [2]. The fishing cat lives mainly in the vicinity of swamps and mangroves. It lives exclusively in South and Southeast Asia and is more prevalent in Thailand [3, 4]. At present, fishing cat populations are threatened as the wetlands where they live have been destroyed over the last decade. As a result, this species is currently considered vulnerable and is on the Red List of the International Union for Conservation of Nature (IUNC) [3]. Due to the loss of their habitat, fishing cat colonies have been migrating and occasionally living in areas where domesticated animals and humans live [5]. This phenomenon of shared environment may lead to increased pathogen spillover between susceptible wild and domestic animals and *vice versa*. Reports of fatal outbreaks in either wild or domestic animals that may have resulted from pathogen spillover have been documented over the past few decades [6–9]. Thus, the transmission dynamics of pathogens from those animals should be examined and needs further investigation.

Carnivore protoparvovirus-1 (CPPV-1), a viral species including mink enteritis virus (MEV), raccoon parvovirus (RPV), feline panleukopenia virus (FPV) and variants of canine parvovirus (CPV), has been recognized as an important pathogen associated with fatal diseases in both wild and domestic *Canidae* and *Felidae* families [10, 11]. FPV, a common pathogen of the *Felidae* species, has been hypothesized to be a common ancestor of CPV [12–14]. CPV frequently infects animals in the *Canidae* family and there is increasing evidence of infection in the *Felidae* family counterpart [9, 15, 16]. Previously, FPV and CPV have been hypothesized to be a host-restricted virus. However, several findings subsequently indicated that some animals in the family *Canidae* might play a role as a reservoir for FPV or *vice versa* [11, 17, 18]. Moreover, several reports have indicated that the FPV and CPV variants (CPV-2a, -2b, and -2c) share susceptible hosts. These hosts are not only domestic animals but also a variety of carnivorous species [15, 19, 20], suggesting a multi-host range of CPPV-1 [11].

From 1996 to 1997, CPV variants were first detected in leopard cats and were initially designated as leopard cat parvovirus (LCPV), suggesting the first description of CPPV-1 in the genus *Prionailurus* [21, 22]. Some of these LCPV strains were revealed to be genetically divergent from the previously described CPV-2a and -2b due to a mutation of the G300A amino acid in the capsid (VP) protein. This resulted in a change in its antigenic properties that is now used as a unique mutation point to differentiate CPV-2c from other CPV variants [22]. Discovery of the new CPV variant (currently known as CPV-2c) in leopard cats supported the idea of co-evolution of CPPV-1 among animal species. Later, infections of FPV and CPV variants were reported in leopard cats [9]. This suggested that other species in the same genus as the leopard cat (genus *Prionailurus*) may be susceptible to CPPV-1 infection. Until now, susceptibility to CPPV-1 infection in other species in the genus *Prionailurus* is still largely unknown. Deciphering how CPPV-1 variants are evolving and how newly emerging variants behave may help elucidate how to prevent a possible outbreak of the new variant. Even if the new virus gains wider host ranges, fatal outbreaks or atypical lesions associated with the infection might be observed in a new, susceptible host. This study describes a fatal CPPV-1 infection in wild fishing cats *(Prionailurus viverrinus)*. Genetic characterization of the obtained CPPV-1 reveals FPV as a causative agent. This finding provides the first evidence of FPV infection in this species, and it unveils the unique cellular tropism of this virus. Further elucidation of FPV infection and its variants in this species is necessary for its account for global infection in numerous vulnerable species.

## Materials and methods

### Animals and routine postmortem examination

In June 2019, two wild fishing cats (*P. viverrinus*), designated case nos. 1 & 2 were found dead in the suburban area in Nakhon Ratchasima Province, Thailand. Later in August 2019, another fishing cat, designated case no. 3, was being rescued from the area where the first two fishing cats were found. Due to the severity of dehydration, case no. 3 died during referral to the animal hospital. These three fishing cats were submitted for postmortem investigation as a routine process. Unfortunately, samples of free-roaming domesticated animals or wild species harbored in the same area were not available; thus, they were not included in this study. Routine postmortem examination was performed on all fishing cats. Selective vital organs including lung, liver, heart, spleen, and kidney were sampled from all fishing cats, while the intestine and mesenteric lymph node were additionally collected from fishing cat no.3. All selected tissues were submitted for histological examination and individually sampled for further molecular assays. For histology, the tissues were immersed in 10% (v/v) neutral buffered formalin for at least 24 hours and routinely processed. Then they were stained with hematoxylin and eosin (H&E) using standard procedures prior to investigation by an ACVP board-certified veterinary pathologist (TK). Fresh tissue samples obtained from the three fishing cats were stored at −80°C for molecular studies. All procedures were performed in accordance with the guidelines and regulations following the approval of the Chulalongkorn University Animal Care and Use Committee (No. 1931036).

### General virological molecular assays

The fresh tissue samples including heart, lung, liver, spleen, and kidney of all fishing cats plus additional intestinal and mesenteric lymph node tissues of fishing cat no. 3, were subjected to viral nucleic acid extraction by individually homogenizing them with 1% (*v/v*) phosphate-buffered saline (PBS). Subsequently, the total viral nucleic acid was extracted using a commercial viral nucleic acid extraction II kit (Geneaid, Taipei, Taiwan) following the manufacturer's suggestion. The extracted nucleic acids were then qualified and quantified by their $A_{260}/A_{280}$ absorbance ratio using the spectrophotometry method (NanoDrop, Thermo Scientific™, Waltham, MA, USA). All extracted nucleic acids were further subjected to viral molecular investigation using polymerase chain reaction (PCR) analysis with several pan-virologic-family PCR testing panels including the detection of the herpesvirus [23], paramyxovirus [24], pneumovirus [25], calicivirus [26], influenza virus [27], bocavirus [26, 28], parvovirus [29] and coronavirus [30, 31]. The PCR protocols, reagents, cycling conditions and positive/negative controls used in the reactions are described in the S1 File. PCR detection specific of the glyceraldehyde-3-phosphate dehydrogenase (GAPDH) gene of feline was used as an internal control as described previously [32]. Subsequently, the positive PCR amplicons were visualized using a QIAxcel capillary electrophoresis platform (Qiagen, Hilden, Germany) as previously described [33]. The positive amplicons of each PCR assay were purified and subjected to bidirectional Sanger sequencing (Macrogen Inc, Seoul, South Korea). Due to moderate autolysis of the tissues, it was not possible to perform bacteriological investigation on these fishing cats.

### *In situ* CPPV-1 detection using immunohistochemistry (IHC), *in situ* hybridization (ISH) and Transmission Electron Microscopy (TEM)

Initial virologic molecular screening revealed positive amplicons of pan-parvovirus PCR and the sequencing results also showed potential DNA sequences of CPPV-1. These findings prompted us to further confirm the presence of CPPV-1 and to localize the distribution of

CPPV-1 in the tissues of fishing cats. Formalin-fixed paraffin-embedded (FFPE) tissues including lung, liver, heart, spleen, kidney and intestine were subjected to CPPV-1 IHC detection using a mouse monoclonal anti-canine parvovirus antibody (ab59832, Abcam, Cambridge, UK) with the horseradish peroxidase (HRP) detection system (EnVision polymer, Dako, Glostrup, Denmark). Briefly, the FFPE sections were cut to 4-μm thickness and further deparaffinized and rehydrated. The slides were then pretreated, endogenous peroxidase blocked and processed as previously described [11]. Sections of the intestinal tissue of a FPV-infected cat and a CPV-infected dog were used as positive controls, while identical sections incubated with purified goat-mouse and rabbit IgGs (IHC universal negative control reagent, code ADI-950-231-0025, Enzo Life Sciences, Farmingdale, NY, USA), were used as negative controls. Regarding the renal tubular necrosis present in the areas where CPPV-1 IHC positive signals were observed, the kidney sections of all fishing cats were subjected to Periodic Acid-Schiff (PAS) special staining, in order to demonstrate the cellular architecture of kidney tissues.

Furthermore, the FFPE kidney sections of fishing cat nos. 1 & 3 were subjected to ISH and TEM analysis in order to support the result of positive CPPV-1 IHC staining and to ultra-structurally demonstrate the viral particles in renal tissues, respectively.

For the ISH, the CPPV-1 DNA probe covering 393 bp [34] of the capsid gene of CPPV-1 was constructed using a PCR DIG Probe Synthesis Kit (Roche Diagnostics, Basel, Switzerland), according to the manufacturer's suggestions. The thermal cycling reaction and condition were performed as previously described [34] with the exception by using the digoxigenin (DIG)-labeled oligonucleotides. The constructed hybridization probe was visualized and confirmed by size resolution on 1.5% (w/v) agarose gel electrophoresis. The ISH with chromogenic DNA was done as previously described [26] with minor modifications. Briefly, after deparaffinization, rehydration, and subsequent rinsing in PBS, the 4-μm-thick FFPE slides were subjected to proteolytic digestion, post-fixation, and pre-hybridization. The slides were then hybridized with the 25 ng/μL ISH probe at 42°C overnight. The hybridization signals were visualized by coupling with 100 μL of anti-DIG-AP Fab fragments (Roche, Basel, Switzerland) (1:200 in 1X Blocking solution) in combination with Liquid Permanent Red (LPR) (Dako, Glostrup, Denmark). Slides were then counterstained with hematoxylin. Red precipitates in the correlation of cellular morphology were thought to be positive for ISH. For negative controls, the tilapia lake virus (TiLV) probes [35] were employed, instead of the CPPV-1 probe.

The TEM samples were prepared in accordance with pop-off techniques with modifications [36–38] and were stained with heavy metals as previously described [39, 40]. The ultrastructure was investigated using TEM (HT7800; Hitachi, Tokyo, Japan) operated at 80 kV.

## Full-length genetic characterization of fishing cat CPPV-1

As the results of virologic PCR and IHC screening were in agreement and the results of partial VP-2 genome sequences obtained from pan-parvoviral PCR suggested genetic similarity between case nos. 1 & 2, but not for case no.3, we further characterized the full-length CPPV-1 sequence from these fishing cats (case nos. 1 & 3) and classified the CPPV-1 origin using a set of degenerated primer pairs specific for FPV and CPV retrieved from a previous publication [26]. Briefly, the extracted nucleic acids obtained from spleen and kidneys of two fishing cats plus additional intestinal tissue of fishing cat no. 3, were individually amplified using a GoTaq® Hot Start Green Master Mix (Promega, Madison, WI, U.S.A.) and specific primers (S1 File). The PCR conditions have been described previously [26]. The target amplicons were visualized using 2% (w/v) agarose gel electrophoresis and further purified using a Monarch DNA Gel extraction kit (New England Biolab, Frankfurt, Germany) prior to commercial

Sanger sequencing. The complete genome sequences of the CPPV-1 isolates obtained from the two fishing cats were deposited in GenBank (accession numbers MW145540-MW145541).

## Phylogenetic, recombination and evolutionary analyses of fishing cat CPPV-1

The full-length genome sequences of the fishing cat CPPV-1 strains were aligned with the published complete CPPV-1 sequences obtained from various host species, available in GenBank using MAFFT V. 7 (http://mafft.cbrc.jp/alignment/server/) and MEGA7 software (http://www.megasoftware.net). The sequence alignments were then subjected to phylogenetic tree construction. The phylogenetic tree was created using the maximum likelihood (ML) method with the Hasegawa–Kishino–Yano with gamma distribution (HKY+G) model, which was selected using the find-best-fit model algorithm in MEGA7 according to the Bayesian information criterion. The tree was bootstrapped with 1,000 replicates. Sequence pairwise similarity among CPPV-1 genomes was calculated using the maximum composite likelihood model and their evolutionary distance was analyzed in MEGA7. The output pairwise nucleotide identities were generated and visualized in Microsoft Excel format (S2 File). The output alignment sequence of CPPV-1 was then used as a template for genetic recombination analysis using two statistically independent software packages. Briefly, the alignment identified the potential recombination strains using integrated Recombination Detection Program 4 (RDP4) Package V. Beta 4.94 software and this was crosschecked with a similarity plot and bootscanning analysis in the SimPlot software package V. 3.5.1. The settings of the programs and interpretation of the results were conducted as previously described [26, 41].

For evolutionary analysis of the fishing cat CPPV-1, a dataset of 224 complete genome sequences containing 26 FPV-, 6 MEV- and 192 CPV sequences, all originating from 18 countries between 1979 and 2019, was retrieved from the GenBank database. Evolutionary analysis was performed with the 2 potential CPPV-1 sequences obtained from fishing cats using the Bayesian Markov chain Monte Carlo (BMCMC) model implemented in BEAST V. 2.4.8 [42]. The jModelTest [43] was performed to identify the best fitting nucleotide substitution model for multiple alignment sequences. The best-fit substitution models under lognormal relaxed and strict clock models at constant population sizes as priors were implemented to account for varied evolutionary rates among lineages. The coalescent Bayesian skyline tree prior and empirical base frequencies were obtained under the best-fit clock model and run for 100 million chains, sampling every 10,000th generation, with the first 10% discarded as burn-in. The convergence of parameters was confirmed by calculating the effective sample size using the TRACER program V. 1.7.0 [44]. The maximum clade credibility trees were annotated using TreeAnnotator V. 1.8.3 [42]. The phylogenetic tree with estimated divergence, variable timeline, posterior probability and 95% highest posterior density (HPD) was generated and displayed using FigTree V. 1.4.3 [45].

## Retrospective study of CPPV-1 antigen in wildlife carnivores

In order to explore the role of CPPV-1 associated with morbidity and mortality with unknown causes in wild carnivores in the distant and recent past, 305 selective fresh samples were included for genomic extraction and identification targeting of the CPPV-1 capsid gene as described above. These samples have been obtained since 2013 as either fecal swabs or intestinal contents. They were obtained from 136 zoo animals from 27 different carnivorous species (S1 Table).

## Results

### Postmortem findings and tissue localization of fishing cat CPPV-1

All necropsied fishing cats were moderately emaciated with varying degrees of dehydration while 2 of the 3 fishing cats (case nos. 1 & 2) showed moderate autolysis. Prominently, the spleen and kidneys were congested in all fishing cats. Fishing cat no.3 had macroscopic lesions of catarrhal enteritis that contained a watery brown-yellowish content in their lumens and congestion of the mesenteric lymph node. Histologically, the lung of fishing cat no. 3 revealed severe congestion with infiltration of mononuclear cells in alveoli. The liver showed mild peri-portal hepatitis, characterized by mononuclear cell infiltration at liver portal triads.

Similar degrees of splenic congestion with few numbers of splenic lymphoid follicles were observed in all fishing cats. The lymphoid follicles were depleted and the remaining lymphoid follicles amid collapsed splenic architecture with increased numbers of prominent splenic trabeculae (Fig 1A). There were scattered karyorrhectic debris of lymphocytes with

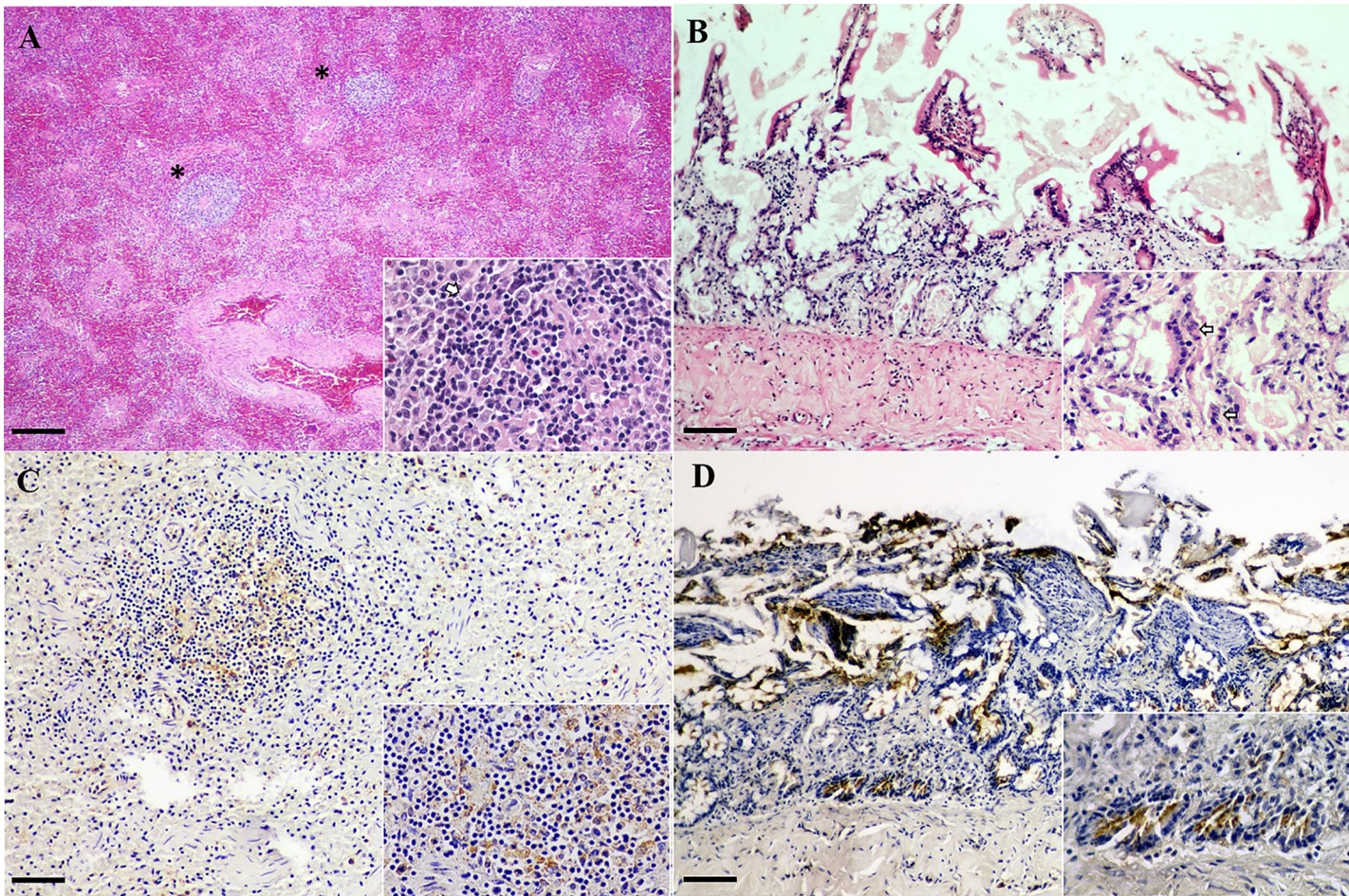

**Fig 1. CPPV-1 infection in fishing cats.** Demonstrative H&E (A, C) and CPPV-1 IHC (B, D) pictures from fishing cats. (A) Fishing cat no. 1. Diffuse congested spleen with sparse numbers of lymphoid follicles. Center of one of the remaining lymphoid follicles contained eosinophilic fibrillar material (fibrin) intermixed with scattered karyorrhectic debris of lymphocytes (lymphocytolysis) (inset). Few numbers of these lymphocytes contained 5–7 μm basophilic intranuclear inclusion bodies that marginated the nuclear chromatin (arrow). (B) Fishing cat no. 3. Shortening of villi with occasional dilated crypts that contained eosinophilic proteinaceous substances and lined by markedly attenuated or necrotic crypt epithelial cells (inset). Many crypt epithelial cells were pyknotic and karyorrhectic and rare cells contained similar basophilic intranuclear inclusion bodies (arrows). (C) Fishing cat no. 1. The CPPV-1 immunoreactivity was frequently observed in the cytoplasm of mononuclear cells in the area of splenic lymphoid follicle. (D) Fishing cat no. 3. CPPV-1 IHC signals were diffusely detected and the immunoreactivity signals were markedly localized in the cytoplasm of cryptal epithelial cells (inset). Bars indicate 25 μm for (A) and 120 μm for (B–D).

accumulations of eosinophilic fibrillar materials in the center of such follicle. Few numbers of these lymphocytes contain 5–7 μm basophilic intranuclear inclusion bodies that marginate the nuclear chromatin.

Shortening of intestinal villi with collapse of mucosal architecture was present in the intestinal section of fishing cat no. 3 (Fig 1B). Some crypts were dilated, and they contained eosinophilic proteinaceous material and were lined by markedly attenuated or necrotic crypt epithelial cells. Many cryptal epithelial cells were pyknotic and karyorrhectic and rare cells contained similar basophilic intranuclear inclusion bodies. CPPV-1 IHC was used to demonstrate the presence of the virus in tissues and to describe the pathological role of CPPV-1 in the lesions observed in routine postmortem examination. The residing mononuclear cells in the spleen revealed positive intense immunoreactivity of CPPV-1 in all fishing cats (Fig 1C). The immunoreactivity was also observed in most of the cytoplasm of cryptal epithelial cells and was compatible with the intestinal lesions of fishing cat no. 3 (Fig 1D).

Notably, various degrees of renal tubular vacuolation and multifocal renal hemorrhage were evident in all fishing cats (Fig 2A). Some renal tubular epithelial cells were hypereosinophilic with nuclear pyknosis, while the rare vacuolated tubular epithelial cells revealed vesiculated nuclei with marginating nuclear chromatin (Fig 2B). Regarding the kidney lesions, we further performed PAS staining to demonstrate the cellular architecture of kidney tissue and to exclude systemic hypoxia as being the possible cause of this lesion. PAS staining demonstrated the renal tubular basement membranes were intact (Fig 2B, inset).

Within the kidney sections, CPPV-1 immunolabelling was strong and frequently seen in the cytoplasm of renal tubular epithelial cells (Fig 2C) and urothelial cells at the renal pelvis in all cases. No evidence of immunogenic reaction was observed in the negative controls (S1 Fig). The atypical CPPV-1 localization in renal tissues revealed by IHC suggested that we should conduct additional investigations to confirm the presence and localization of the CPPV-1 in kidney using the ISH and TEM with pop-off technique. The CPPV-1 ISH-immunoreactivity was positive in all fishing cats and diffusely localized in the nucleus of renal tubular epithelial cells, where the tubular lesions were observed (Fig 2D). No reaction was observed in the negative controls (S1 Fig.)

In accordance with results of positive immunological signals of ISH that localized in the nucleus of renal epithelial cells, numerous small electron-dense viral particles, estimated at about 19–22 nm in diameter, were observed. These were clustered in the nuclei of these renal tubular and urothelial cells (Fig 3).

## Detection of fishing cat CPPV-1 and genomic analysis

Primarily, the pan-family PCR panels specific for various viruses including herpesvirus, paramyxovirus, pneumovirus, calicivirus, influenza virus, bocavirus, coronavirus and parvovirus were tested on the extracted viral nucleic acid samples. This revealed positive results only with the pan-parvovirus PCR in intestine, lymph node and kidney samples of the three fishing cats. Conversely, other pan-PCRs for the other virus detections were negative in all tested tissue samples.

From previously obtained results of pan-parvovirus PCR positivity in all fishing cats, we further characterized the coding genome sequences in two fishing cats using several primer pairs in the extracted nucleic acid samples obtained from intestine (case no.1) and kidney (case no.3). The 4,462 and 4,430 base pairs of complete coding genomes of two fishing cats CPPV-1 strains were detected and were designated strain 19ZP004-TH/2019 (case no.1, accession no. MW145540) and 19ZP005-TH/2019 (case no.3, accession no. MW145541), respectively. After genetic analysis, the complete coding genomes revealed genetically similar results

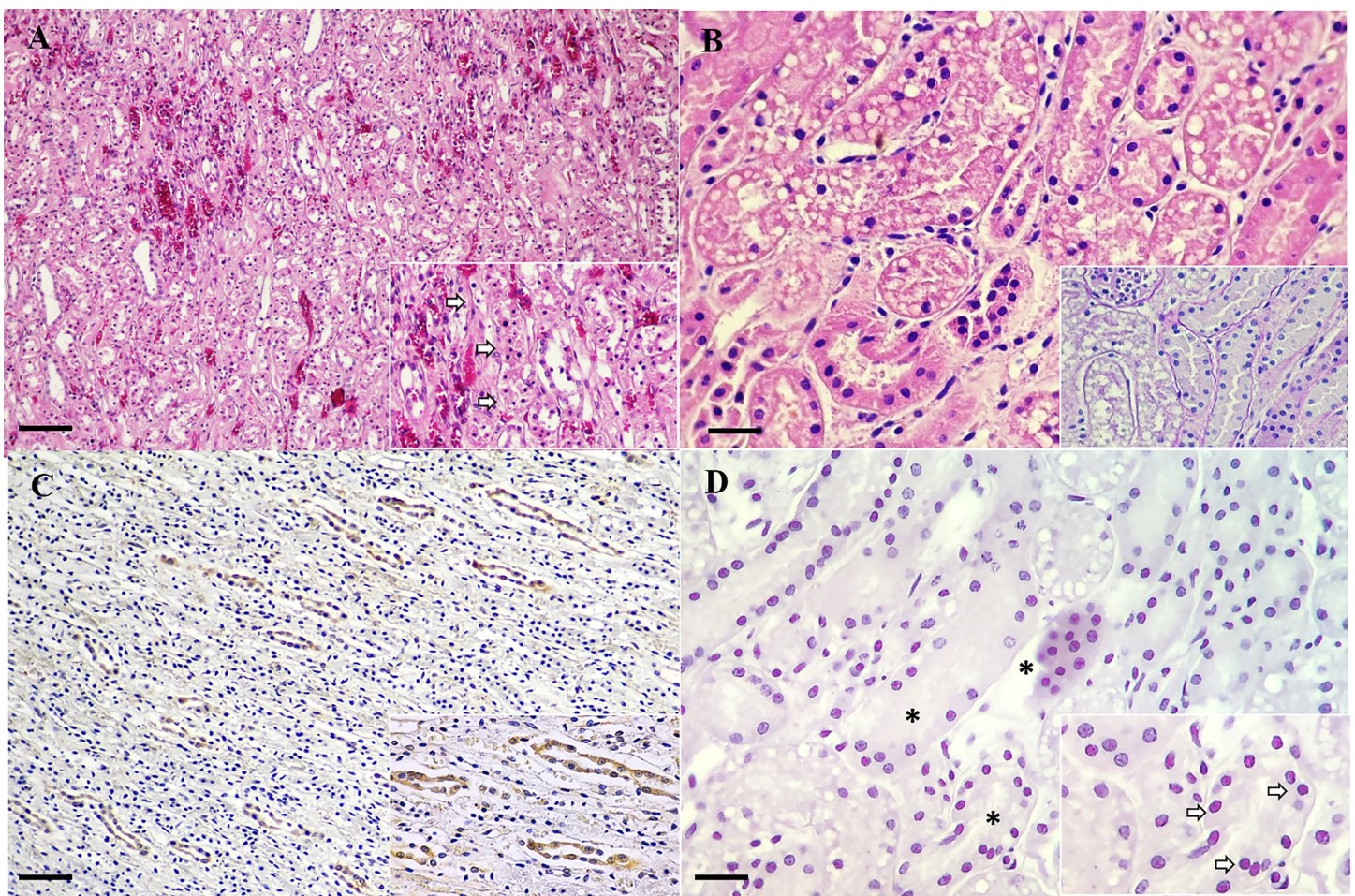

**Fig 2. CPPV-1 infection in fishing cats.** Demonstrative H&E **(A)**, PAS staining **(B)**, CPPV-1 IHC **(C)** and *in situ* hybridization (ISH) **(D)** photomicrographs of kidney from fishing cats. **(A, B)** Fishing cat no.3. **(A)** Multifocal renal hemorrhage with renal tubular vacuolation. Few tubular epithelial cells were hypereosinophilic with pyknotic nuclei (inset, arrows). **(B)** Renal tubular epithelial cells exhibited cytoplasmic vacuole and the nuclei of rare epithelial cells are vesiculate with marginating nuclear chromatin. PAS staining highlighted intact basement membrane (inset). **(C)** Fishing cat no. 2. CPPV-1 IHC-immunoreactivity (dark-brown color) was diffusely observed in the renal tubules and frequently detected in the cytoplasm of renal tubular epithelium (inset). **(D)** Fishing cat no.1. CPPV-1 ISH-immunoreactivity (reddish pink color) was abundantly observed in renal tubules (asterisks), and it localized in the nuclei of renal tubular epitheliums (inset, arrows). Bars indicate 50 μm for (A, C) and 120 μm for (B, D).

among CPPV-1 strains obtained from fishing cats by showing that these fishing cat CPPV-1 were FPV. Deduced amino acid comparison between the CPPV-1 variants and the fishing cats FPV are described in Table 1. Of note, the fishing cat FPV revealed unique amino acid mutations of I566M and M569R in the structural (VP1 & 2) protein, that were not detected in previous CPPV-1 variants.

## Phylogenetic and evolutionary analysis

For comparing the genetic relationship between detected fishing cat CPPV-1 and other previously published CPPV-1 genomes, phylogenetic analysis based on complete coding genome sequence was conducted. The ML phylogenetic tree showed that most FPV, MEV and CPV formed a very distinct clade among groups (Fig 4). The two CPPV-1 genomes from the studied fishing cats (indicated by red triangles) were clustered together with the same lineage of previously published FPV sequences obtained from various hosts, confirming that the obtained

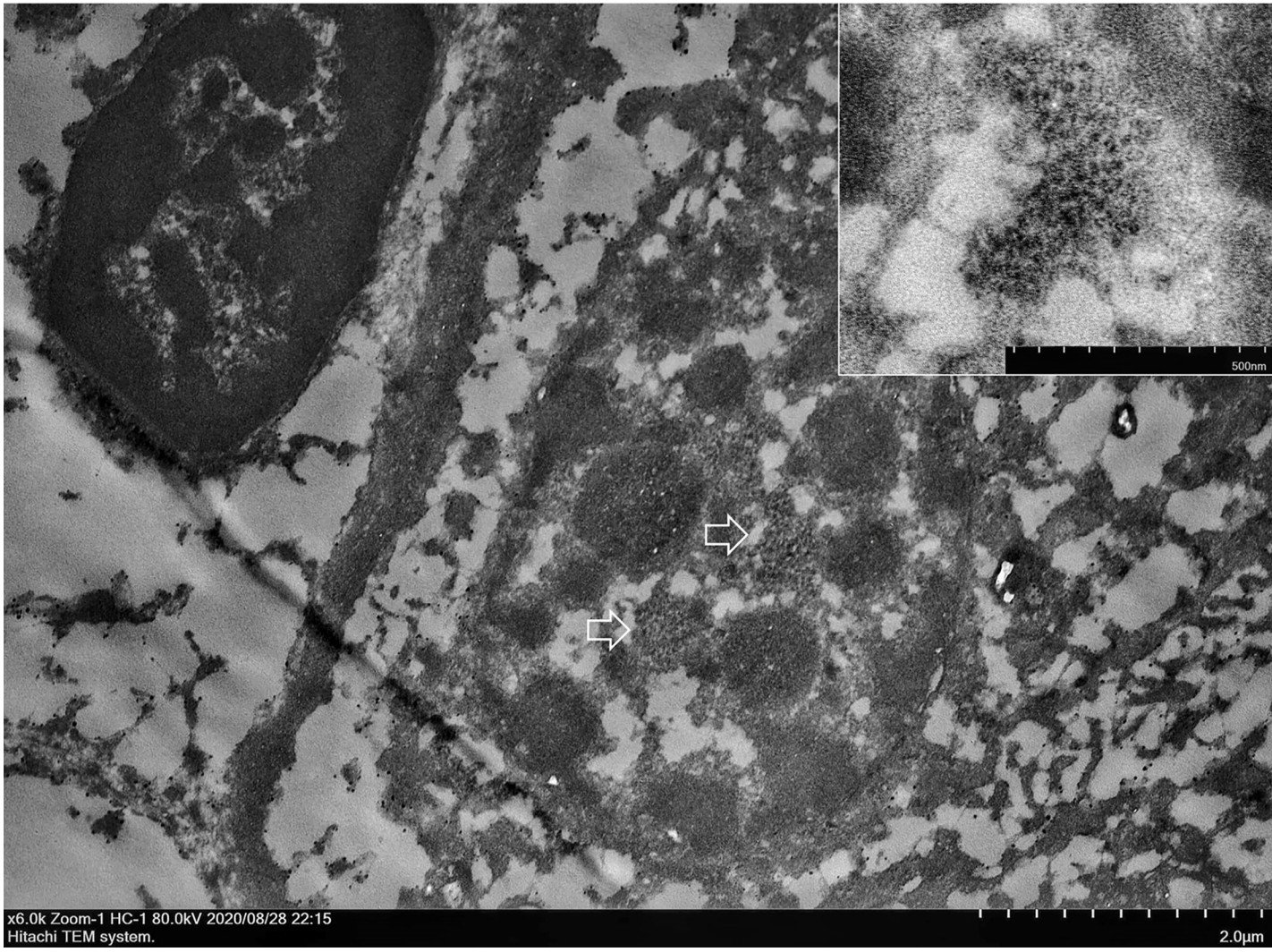

**Fig 3. CPPV-1 viral particles in the nucleus of renal tubular epithelial cells.** Transmission electron microscopy (TEM) using the pop-off technique. Ultrastructural demonstration of clustering of electron-dense particles (arrows). The icosahedral particle size was 19–22 nm diameter located in the nucleus of renal tubular epithelial cells (inset). Scale bars as shown in the figure.

fishing cat CPPV-1s were FPV. The detected genomes were clustered together and most closely related to the FPV genome detected in domestic cats in Thailand (accession no. MN127780). To further elucidate the evolutionary process of detected fishing cat CPPV-1, several recombination detections and evolutionary analyses on a dataset of 224 CPPV-1 complete genome sequences were conducted. The recombination analysis revealed that no recombination events were found in fishing cat CPPV-1 sequences.

The evolutionary tree of these data showed two main distinct separated clades (Fig 5). One clade included FPV and MEV and another clade included all CPV sequences. The overall evolutionary rate was estimated at $1.08 \times 10^{-4}$ substitutions/site/year (95% HPD: 1.95–$2.87 \times 10^{-4}$). The evolutionary tree showed that fishing cat FPV sequences were clustered in the same lineage of FPV as that isolated from Thailand. They have shared genetic evolution with FPV genomes detected in domestic cats in China and may have the same origin since around 1970.

**Table 1. Alignment of capsid amino acid sequence of CPPV-1 isolated from fishing cats with previously described CPPV-1 variants retrieved from the NCBI database.** The accession numbers, the CPPV-1 variant, infected host, country and year of sample collection are presented.

| Isolations | Amino acid position of VP2 protein | | | | | | | | | | | | | | | |
|---|---|---|---|---|---|---|---|---|---|---|---|---|---|---|---|---|
| | 80 | 87 | 93 | 101 | 103 | 232 | 297 | 300 | 305 | 323 | 370 | 426 | 564 | 566 | 568 | 569 |
| MN908257 FPV Tiger China 2019 | K | M | K | T | V | V | S | A | D | D | Q | N | N | I | A | M |
| MG764511 FPV Lion China 2015 | . | . | . | . | . | . | . | . | . | . | . | . | . | . | . | . |
| MN862743 FPV Mink Canada 2018 | . | . | . | . | . | . | . | . | . | . | . | . | . | . | . | . |
| MF069447 FPV Raccoon Canada 2016 | . | . | . | . | . | . | . | . | . | . | . | . | . | . | . | . |
| MN862748 FPV River Otter Canada 2019 | . | . | . | . | . | . | . | . | . | . | . | . | . | . | . | . |
| MN862744 FPV Pine Marten Canada 2016 | . | . | . | . | . | . | . | . | . | . | . | . | . | . | . | . |
| MN451652 FPV Arctic Fox Finland 1983 | . | . | . | . | . | . | . | V | . | . | . | . | . | . | . | . |
| MH127910 FPV Leopard Cat Taiwan 2017 | n/a | n/a | n/a | n/a | n/a | n/a | n/a | . | . | . | . | . | . | . | . | . |
| MH127911 FPV Leopard Cat Taiwan 2017 | n/a | n/a | n/a | n/a | n/a | n/a | n/a | . | . | . | . | . | . | . | . | . |
| Fishing cat CPPV-1/ZP004 | . | . | . | . | . | . | . | . | . | . | . | . | . | M* | . | R* |
| Fishing cat CPPV-1/ZP005 | . | . | . | . | . | . | . | . | . | . | . | . | . | M* | . | R* |
| MG924893 FPV Cat China 2016 | . | . | . | . | . | . | . | . | . | . | . | . | . | . | . | . |
| MN127779 FPV Cat Thailand 2018 | . | . | . | . | . | . | . | . | . | . | . | . | . | . | . | . |
| MN127781 FPV Cat Thailand 2019 | . | . | . | . | . | I | . | . | . | . | . | . | . | . | . | . |
| KP019621 Civet CPPV-1 Thailand 2013 | . | . | . | . | . | . | . | . | . | . | . | . | . | . | . | . |
| KR002804.1 CPV 2a Dog China 2014 | R | L | N | | A | I | A | G | Y | N | . | | S | . | G | . |
| KR002799.1 CPV 2b Dog China 2013 | R | L | N | | A | I | A | G | Y | N | . | D | S | . | G | . |
| MF177228.1 CPV 2c Dog Italy 2009 | R | L | N | | A | I | A | G | Y | N | . | E | S | . | G | . |
| MN747143.1 MEV Mink China 2018 | R | . | N | I | A | I | . | D | . | N | . | | S | . | G | . |

CPPV-1: carnivore protoparvovirus-1; CPV: canine parvovirus; FPV: feline parvovirus; MEV: mink enteritis virus.

*Unique amino acid mutations observed in fishing cat CPPV-1; n/a: not available data.

## Retrospective study of CPPV-1 in other wild samples

A retrospective study of 136 zoo-wild animals derived from 27 different carnivores did not reveal the presence of CPPV-1 genomic antigen as detected by PCR.

## Discussion

Circulation of CPPV-1 variants that includes CPV and FPV have been identified in both domestic and wild carnivores. While fatal infection of CPPV-1 in domestic dogs and cats is common, this infection is sporadically seen in wild carnivores [46]. Several studies have indicated that many wild carnivores are susceptible to CPPV-1 infection including wild felids, minks, otters, skunks, raccoons, foxes, wolfs, coyotes, civets and martens [11, 19]. Particular to the genus *Prionailurus*, only leopard cats have been reported to be infected with CPPV-1 variants [9]. In this study, we describe the natural infection of FPV, a member of CPPV-1, in wild fishing cats (*Prionailurus viverrinus*). This is the first report with a description of distinct viral distribution and tropism in kidney tissues as confirmed via antigenic detection using PCR, IHC, ISH and TEM analysis of the ultra-structure.

Cross-species transmission of CPPV-1 variants between domestic and wild carnivores has been observed, and some fatal CPPV-1 outbreaks have been speculated to be associated with pathogen spillover among habitats [9, 11]. Due to the current loss of wildlife habitat, wild colonies have been migrating and sharing habitats with domestic animals, which has led to increased pathogen spillover between susceptible animals. In this study, we found that the fishing cat CPPV-1 genomes have shared an evolutionary pattern with the FPV strains detected in

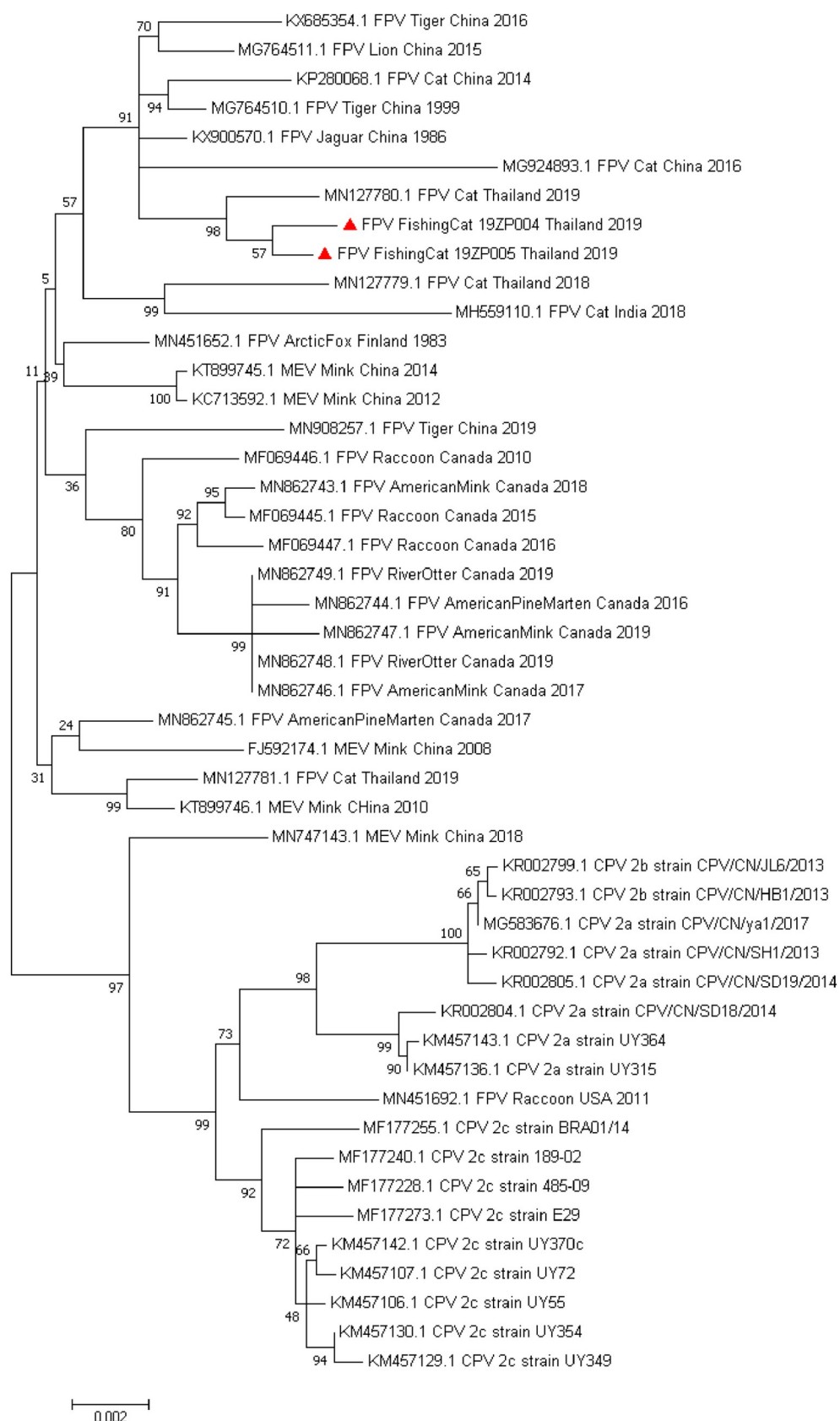

**Fig 4. Phylogenetic topology revealing the genetic relationship of the full-length coding sequences between the fishing cat CPPV-1 and other CPPV-1 variants.** The phylogenetic tree showed that the fishing cat CPPV-1 sequences (indicated by red triangles) were clustered together with the FPV genomes with the genome most related to the FPV genome detected in the Thai (MN127780). The GenBank accession numbers of previously described CPPV-1 sequences used in this analysis were indicated.

domestic cats in Thailand. These results show a similar evolutionary pattern to the FPV strains from common ancestors isolated from domestic cats in China since the 1970s. This result may imply either possible evidence of cross-species transmission between domestic cats and fishing cats or co-evolution of the CPPV-1 variants. However, owing to the limitation of samples obtained from free-roaming animals in the same habitats and the small numbers of investigated fishing cats, the primary reservoir of the CPPV-1 in this study could not be identified and further investigation is needed.

Although several studies have indicated that various amino acid residues in the capsid gene are associated with both the ability to infect new hosts and the antigenicity of CPPV-1 [18, 47], the single amino acid mutation at position 300 of the capsid protein (VP2) has been reported as a key determinant for multiple host ranges [48]. In this study, the CPPV-1 variant obtained from the fishing cats revealed the same amino acid patterns as found as in FPV sequences, with the presence of alanine (A) as a key amino acid determinant at capsid position 300. While the fishing cat capsid protein of CPPV-1 revealed asparagine (N) and alanine (A) amino acids located at positions 564 and 568, respectively (which are considered to be critical for FPV replication in the feline host), they had 2 unique amino acid mutations at positions I566M and M569R. These mutations can differentiate this virus from previously described CPPV-1 variants. Previous analysis of the ancestral reconstruction found that FPV can infect distinct carnivores with few changes in the VP2 gene [49]. Thus, the FPV-like parvovirus was tentatively named as a potentially novel virus detected in the fishing cats and this may support the first identification of this virus in this species. However, a small number of investigated fishing cats compared to a small number of other sequences may not give a definitive interpretation. This needs further study for more definite conclusions to be drawn.

Fishing cat CPPV-1 localizations were observed in various tissues including intestinal epithelial cells and lymphoid cells, where the most common tissue localization of the CPPV-1 infection is found in other species [20, 26, 50, 51]. Interestingly, since initial investigation of viral genomic detection using PCR and viral localization using IHC revealed distinct viral localization in kidney tissues, we performed PAS staining to demonstrate the cellular architecture of renal tissue and to exclude systemic hypoxia as being the possible cause of tubular necrosis. PAS staining revealed the most tubular basement membranes were intact, suggesting that this lesion may not result from systemic hypoxia [52]. However, other causes such as toxin-induced tubular necrosis can cause this lesion. Thus, the role of FPV-like parvovirus infection associated with kidney lesions still needs further investigation. Moreover, we attempted to further elucidate extant localization of the virus in kidney tissue using ISH and TEM with the pop-off technique [37] in the areas where CPPV-1 IHC-immunoreactivity was detected. The electron-dense particles were observed in the nucleus of renal tubular epithelial cells and urothelial cells, supporting the positive *in situ* immunoreactivity of CPPV-1 in kidney tissue. FPV-infected cats shed the virus predominantly in feces, but the presence of active virus in urine has been described [53, 54]. Recent studies have indicated that parvovirus infection has a role in renal failure [55]. However, the localization of FPV and the CPV counterpart in the kidney tissue has not previously been investigated. Furthermore, a mouse kidney parvovirus (MKPV) has been proven to be a nephrotropic virus by showing an affinity to infect and localize in renal tubular epithelial cells, resulting in kidney diseases [56, 57].

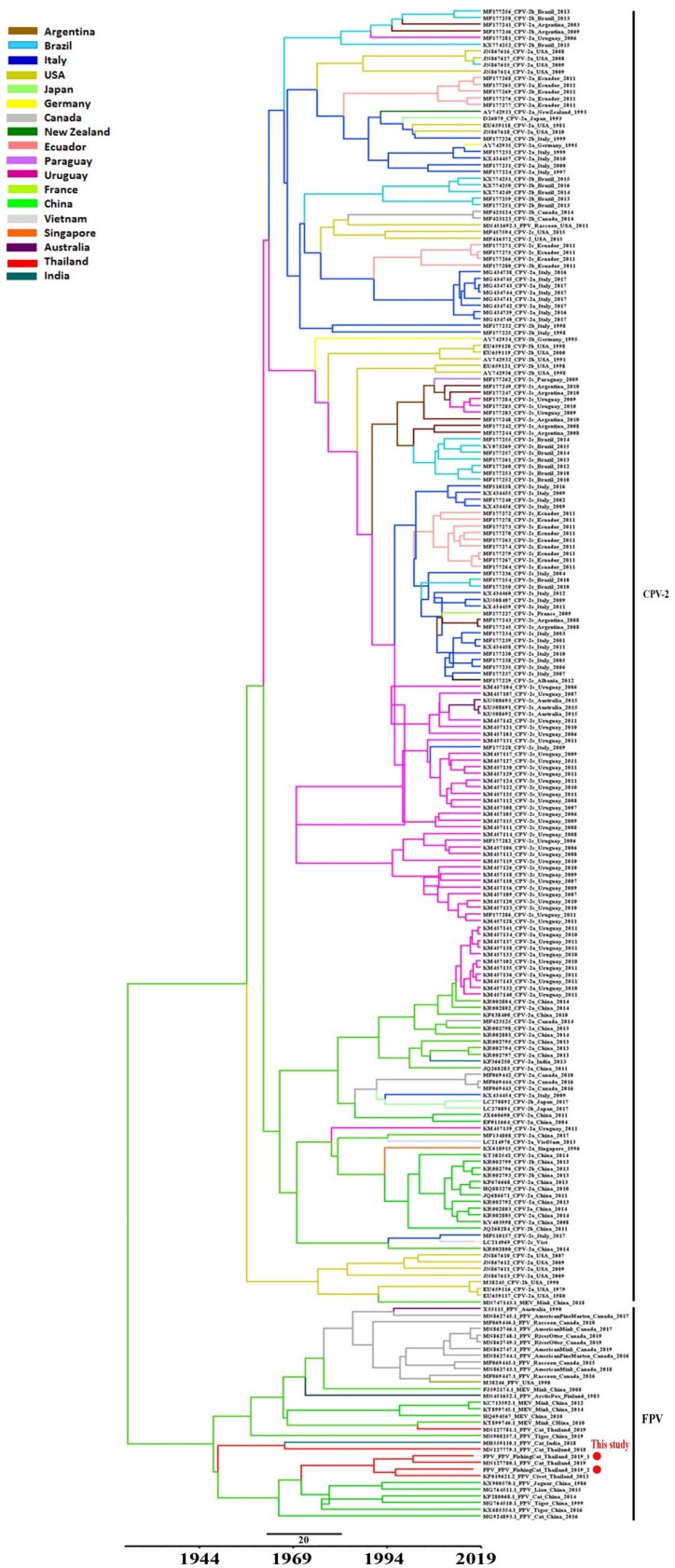

**Fig 5. Evolutionary analysis of the fishing cat CPPV-1 with other CPPV-1 variants.** The fishing cats CPPV-1 genomes have shared their evolution with the FPV detected in Thai cat populations and originated from the same ancestor as the FPV genomes detected in China since the 1970s.

Based on our knowledge, CPPV-1 localization in the kidneys has not been previously described. This suggests the novel description of a unique viral localization in this species or even if it can be found in FPV-infected cats. This finding may support previous studies which found FPV in the urine. Unique pathological findings may be subjective and may be observed as infection in individuals. Thus, a large-scale investigation of fishing cat CPPV-1 will facilitate a definitive interpretation. In addition, specimens that were submitted mostly from dead animals and had already undergone postmortem autolysis are not suitable for further organ collection from these animals. Thus we were not able to determine the viral localization in other organs. Elucidation of atypical viral localization of fishing cat CPPV-1 would be a useful investigation.

CPPV-1 has been identified in various wild carnivores and it is associated with fatal outbreaks. The negative evidence of retrospective CPPV-1 detection in zoo-wild samples in this study may result from either no close-contact among susceptible animals or, on the other hand, prolonged sample storage that affects the stability of genomic materials. The novel identification of CPPV-1 associated with mortal infection of fishing cats in this study raises concerns about a novel, potentially virulent disease in this vulnerable animal. Therefore, further study of the transmission of CPPV-1 in fishing cats is essential to prevent a loss of other susceptible animal species. Accordingly, prevention strategies should be emphasized in order to manage CPPV-1 infections, not only in this species but also in free-roaming domestic animals that may serve as potential hosts for this virus.

## Supporting information

**S1 Fig.** Photomicrograph of negative controls for CPPV-1 IHC (A-C) and CPPV-1 ISH (D). No CPPV-1 IHC reaction is present within the negative control section of (A) intestine, (B) spleen, and (C) kidney. (D) No immunoreactivity is present within a kidney section incubating with the TiLV probe (non-related probe).
(TIF)

**S1 Table. List of wildlife carnivore species enrolled in the retrospective study of CPPV-1 detection.**
(DOCX)

**S1 File. Amplification procedures used for routine pan-virologic-family detection and full-length genetic characterization of fishing cat CPPV-1.**
(DOCX)

**S2 File. Pairwise nucleotide distances of fishing cat CPPV-1 and previously published CPPV-1 genomes.** The nucleotide similarity of complete coding sequences of various CPPV-1 variants is compared with the nucleotide sequences of CPPV-1 obtained from fishing cats in this study. Accession nos. are indicated.
(XLS)

## Acknowledgments

We would like to thank Mr. Poowadon Chai-in, for development of sample preparation techniques for TEM, and Dr. Robert D. Gaffin and Dr. Ariyaporn T. Gaffin for proofreading.

## Author Contributions

**Conceptualization:** Chutchai Piewbang, Somporn Techangamsuwan.

**Data curation:** Chutchai Piewbang.

**Formal analysis:** Chutchai Piewbang, Sabrina Wahyu Wardhani.

**Funding acquisition:** Suwimon Boonrungsiman, Nattika Saengkrit, Yong Poovorawan, Somporn Techangamsuwan.

**Investigation:** Chutchai Piewbang, Sabrina Wahyu Wardhani, Jira Chanseanroj, Jakarwan Yostawonkul, Piyaporn Kongmakee, Wijit Banlunara, Somporn Techangamsuwan.

**Methodology:** Chutchai Piewbang, Jira Chanseanroj.

**Project administration:** Chutchai Piewbang, Suwimon Boonrungsiman, Yong Poovorawan, Somporn Techangamsuwan.

**Resources:** Suwimon Boonrungsiman, Nattika Saengkrit, Yong Poovorawan, Somporn Techangamsuwan.

**Software:** Jira Chanseanroj.

**Supervision:** Somporn Techangamsuwan.

**Validation:** Chutchai Piewbang, Tanit Kasantikul.

**Visualization:** Chutchai Piewbang, Tanit Kasantikul.

**Writing – original draft:** Chutchai Piewbang.

**Writing – review & editing:** Chutchai Piewbang, Somporn Techangamsuwan.

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
