## [Decision Letter · Decision Letter 0]

23 Nov 2020

PONE-D-20-33818

Natural infection of parvovirus in wild fishing cats (Prionailurus viverrinus) reveals extant viral localization in kidneys

PLOS ONE

Dear Dr. Piewbang,

Thank you for submitting your manuscript to PLOS ONE. After careful consideration, we feel that it has merit but does not fully meet PLOS ONE’s publication criteria as it currently stands. Therefore, we invite you to submit a revised version of the manuscript that addresses the points raised during the review process.

Many thanks for submitting your manuscript to PLOS One

It was reviewed by two experts in the field who have suggested some modifications be made prior to acceptance

If you could make these modifications, and write a response to reviewers, that will expedite revision upon resubmission

I wish you the best of luck with your revisions

Hope you are keeping safe and well in these difficult times

We look forward to receiving your revised manuscript.

Kind regards,

Simon Clegg, PhD

Academic Editor

PLOS ONE

2. To comply with PLOS ONE submissions requirements, please provide methods of sacrifice in the Methods section of your manuscript.

Reviewers' comments:

Reviewer's Responses to Questions

**Comments to the Author**

1. Is the manuscript technically sound, and do the data support the conclusions?

Reviewer #1: Partly

Reviewer #2: Yes

2. Has the statistical analysis been performed appropriately and rigorously? 

Reviewer #1: I Don't Know

Reviewer #2: N/A

3. Have the authors made all data underlying the findings in their manuscript fully available?

Reviewer #1: Yes

Reviewer #2: Yes

4. Is the manuscript presented in an intelligible fashion and written in standard English?

Reviewer #1: Yes

Reviewer #2: Yes

5. Review Comments to the Author

Reviewer #1: The authors investigated three cases of natural canine ürptpüarvovirus-1 (CPPV-1) infection in wild fishing cats. In their article they describe a new tropism of parvovirus to renal tissue by using PCR, immunohistochemistry and transmission electron microscopy.

Major comments:

1. In the materials and methods (Line 116ff) the authors describe that they isolate nucleic acids from fresh samples. Since the authors also claim a high degree of autolysis in cases 1&2 they have to specify: 1.) nucleic acids were isolated from which organs in which cats, and 2.) PCRs for virus familys were performed on which organs from which cats.

2. The authors describe that they perform pan-virologic-family PCRs - they should specify for which families they used PCR and for which RT-PCR.

3. In addition, the used PCR/RT-PCR kits have to be specified including PCR conditions and applied positive controls for PCR/RT-PCR.

4. Where process controls included during nucleic acid extraction since autolysis may have a significant impact on the performance of the PCR reaction.

5. Regarding the postmortem findings (lines 214ff) the authors stated that "Other organs showed advanced autolysis and so they could not be investigated." Do the authors mean by histology, immunohistochemistry, PCR, TEM? Please specify.

In the following sentence the authors state that histologically all fishing cats showed desquamative enteritis, lymphoid depletion in lymph nodes and spleen. What does this mean in the context of the sentence before? Please include a picture detailing lymphoid depletion in spleen and lymph nodes. Were these findings associated with a positive signal in PCR and/or IHC for CPPV-1?

6. Line 230f: What do you mean with severely collapsed intestinal mucosa?

7. Regarding the tubular necrosis: Do the authors perform PAS-stain to exclude hypoxia as being the cause of the tubular necrosis? Were the necrotic cells positive for CPPV-1?

8. The IHC-picture of the kidney looks like as if almost all renal tubular cells were positive for CPPV-1 antigen. To confirm their finding the authors should perform in situ hybridizytion to show a correlation/specificity of the IHC for CPPV-1.

9. Regarding the retrospective study (lines 315ff): What do the authors mean by "A retrospective study of 136 zoo-wild animals in 27 carnivores revealed the presence of CPPV-1 genomic antigen."? Does this mean 27 cases were positive? The authors should explain which organs were investigated in which animal and what methods including nucleic acid extraction and PCR method were used. Do the authors also perform IHC? If all samples were negative, how can the authors show that this result is not related to tissue storage / fixation?

Minor comments:

Line 90 "... CPPV-1 reveals FPV is a ...": please change is to as

Line 327 "... and tropism and in kidney...": please delete the second "and".

Reviewer #2: Carnivore protoparvovirus-1 (CPPV-1), include feline panleukopenia virus (FPV) and canine parvovirus (CPV), which are widespred among domestic and wild carnivores, causing systemic fatal diseases. Wild fishing cats (Prionailurus viverrinus), is a vulnerable species. Virological (PCR and TEM) and gross and microscopic investigations, identified the opresence of an FPV-like parvovirus in fishing cats found dead. Postmortem examination revealed severe enteritis, lymphadenopathy and nephritis. On whole genome sequencing, the virus closely resembled FPV sequences with two peculiar amino acid mutations I566M and M569R in the capsid protein.

The manuscript is of relevant scientific interest. The introduction is well written. The materials and methods seem adequate. The discussion is correct and rather balanced. I have only minor comments.

i) there is a confusion with the terms CPPV-1 and FPV, that are nearly the same thing. I would suggest to use consistently the term FPV-like parvovirus

ii) there are some parts of the manuscript that could be deleted, reworded or re-phrased.

Abstract: check English. Shorten the final part, very generic.

Line 81-83 it is not clear

Lines 81-88: rephrase

Line 90

Line 316

Line 322: please delete/replace the semicoma (;) after infection

Line 325: FPV is not a variant of CPPV-1

Line 346-347: I would delete the comments (Suggesting… Therefore….)

Line 360-361: rephrase

Line 363: odd sentence (Supporting viral genomic detection…). Rephrase

iii) A table with the list (and details) of primers used (consensus and specific), even as supplemental file, should be added.

iv) General comment: the English is good but should be further refined.

6. PLOS authors have the option to publish the peer review history of their article (what does this mean?). If published, this will include your full peer review and any attached files.

Reviewer #1: No

Reviewer #2: No

---

## [Author Response · Author response to Decision Letter 0]

8 Dec 2020

Rebuttal response on PONE-D-20-33818 

Response: We have revised the manuscript following PLOS ONE's style requirements.

2. To comply with PLOS ONE submissions requirements, please provide methods of sacrifice in the Methods section of your manuscript.

Response: The three fishing cats that enrolled in this study were naturally died by first 2 fishing cats were found dead and another fishing cats died during referral. Thus, we have no information regarding the method of sacrifice. Details of animals in this study were described in Materials and Methods section. 

 Reviewers' comments:

Reviewer's Responses to Questions

Comments to the Author

1. Is the manuscript technically sound, and do the data support the conclusions?

Reviewer #1: Partly

Reviewer #2: Yes

2. Has the statistical analysis been performed appropriately and rigorously?

Reviewer #1: I Don't Know

Reviewer #2: N/A

3. Have the authors made all data underlying the findings in their manuscript fully available?

Reviewer #1: Yes

Reviewer #2: Yes

4. Is the manuscript presented in an intelligible fashion and written in standard English?

Reviewer #1: Yes

Reviewer #2: Yes

5. Review Comments to the Author

Reviewer #1: The authors investigated three cases of natural canine carnivore protoparvovirus-1 (CPPV-1) infection in wild fishing cats. In their article they describe a new tropism of parvovirus to renal tissue by using PCR, immunohistochemistry and transmission electron microscopy.

Major comments:

1. In the materials and methods (Line 116ff) the authors describe that they isolate nucleic acids from fresh samples. Since the authors also claim a high degree of autolysis in cases 1&2 they have to specify: 1.) nucleic acids were isolated from which organs in which cats, and 2.) PCRs for virus families were performed on which organs from which cats.

Response: With the request from reviewer, we have provided additional information of the sample collection by describing details as “The fresh tissue samples including heart, lung, liver, spleen, and kidney of all fishing cats plus additional intestinal and mesenteric lymph node tissues of fishing cat no. 3, were subjected to viral nucleic acid extraction.” Page 6, lines 113-115. For additional information of PCRs for virus families, we have added more details regarding the PCR protocols in S1_File. 

2. The authors describe that they perform pan-virologic-family PCRs - they should specify for which families they used PCR and for which RT-PCR. 

Response: We have added more details in S1_file. 

3. In addition, the used PCR/RT-PCR kits have to be specified including PCR conditions and applied positive controls for PCR/RT-PCR.

Response: We have provided more details as description in S1_file. 

4. Where process controls included during nucleic acid extraction since autolysis may have a significant impact on the performance of the PCR reaction.

Response: Thank you for your concern. At this point, we do respect that the autolysis tissues may significantly impact on the performance of the PCRs, thus the tissue samples which revealed moderate to marked autolysis (presented on necropsy desk) would be not included for the molecular study. Furthermore, we have validated all extracted nucleic acids by further performing the PCR to detect the glyceraldehyde-3-phosphate dehydrogenase (GAPDH) gene as an internal housekeeping gene in order to validate the performance of samples and extraction process. Thus, we have provided the information as “The PCR protocols, reagents, cycling conditions and positive/ negative controls used in the reactions are described in the S1 File. PCR detection specific of the glyceraldehyde-3-phosphate dehydrogenase (GAPDH) gene was used as an internal control as described previously (31).” In page 6, lines 124-127.

5. Regarding the postmortem findings (lines 214ff) the authors stated that "Other organs showed advanced autolysis and so they could not be investigated." Do the authors mean by histology, immunohistochemistry, PCR, TEM? Please specify. In the following sentence the authors state that histologically all fishing cats showed desquamative enteritis, lymphoid depletion in lymph nodes and spleen. What does this mean in the context of the sentence before? Please include a picture detailing lymphoid depletion in spleen and lymph nodes. Were these findings associated with a positive signal in PCR and/or IHC for CPPV-1?

Response: Thank you for question regarding the postmortem findings that the presented statements may lead confusion. Due to the fact that fishing cats no. 1&2 have been found dead and the most of organs revealed autolysis, which is presenting on necropsy desk, resulting in other organs could not be investigated. Thus, we do delete “Other organs showed advanced autolysis and so they could not be investigated” and revise the statement regarding the post-mortem findings as “All necropsied fishing cats were moderately emaciated with varying degrees of dehydration while 2 of the 3 fishing cats (case nos. 1 and 2) showed moderate autolysis. Prominently, the spleen and kidneys were congested in all fishing cats. Note that fishing cat no.3 had macroscopic lesions of catarrhal enteritis that contained a watery brown-yellowish content in their lumens and congestion of the mesenteric lymph node.” in page 11, lines 236-240. Furthermore, we have provided the H&E and CPPV-1 IHC pictures of lymph node as your requested in Fig 1 and the IHC-positive signals were identified in the area of lymphoid necrosis as described in Fig 1 legend.

6. Line 230f: What do you mean with severely collapsed intestinal mucosa?

Response: We have revised the sentence as “(B) Fishing cat no. 3. The intestinal mucosa is diffusely and severely desquamated, the intestinal villi are mostly short, many crypts are lost or contain karyorrhectic debris (inset) and there are decreased numbers of goblet cells.”. Fig1 legend. (Page 12, lines 254-256.)

7. Regarding the tubular necrosis: Do the authors perform PAS-stain to exclude hypoxia as being the cause of the tubular necrosis? Were the necrotic cells positive for CPPV-1?

Response: We have performed the PAS staining in kidney section of all fishing cats. The results were shown in Fig 2B. and described as “Regarding the severe renal tubular necrosis, we further performed PAS staining in kidney sections to demonstrate the cellular architecture of kidney tissue and to exclude systemic hypoxia as being the possible cause of this lesion. PAS staining demonstrated the disruption of renal tubular epitheliums while the architectures of most renal tubular basement membranes were intact (Fig 2B).” in Result section (pages 12-13, lines 264-268) and “we performed PAS staining to demonstrate the cellular architecture of renal tissue and to exclude systemic hypoxia as being the possible cause of tubular necrosis. PAS staining revealed disruption of tubular epithelium while most tubular basement membranes were intact, suggesting that this lesion may not result from systemic hypoxia (51).” in Discussion section (Page 21, lines 406-409). Of note. The CPPV-1 IHC signals were frequently seen in the area of tubular necrosis as we have revised as “The CPPV-1 ISH-immunoreactivity was positive in all fishing cats and diffusely localized in the nucleus of renal tubular epithelial cells, where the tubular necrosis lesions were observed (Fig 2D).” in Results section (page 15, lines 275-277). 

8. The IHC-picture of the kidney looks like as if almost all renal tubular cells were positive for CPPV-1 antigen. To confirm their finding the authors should perform in situ hybridization to show a correlation/specificity of the IHC for CPPV-1.

Response: We performed the ISH in kidney sections, derived from all fishing cats and the result picture was shown in Fig 2D. Details regarding the protocols and results were described in Materials and Methods section (pages 7-8, lines 153-171) and Results section (page 13, lines 275-278, respectively. 

9. Regarding the retrospective study (lines 315ff): What do the authors mean by "A retrospective study of 136 zoo-wild animals in 27 carnivores revealed the presence of CPPV-1 genomic antigen."? Does this mean 27 cases were positive? The authors should explain which organs were investigated in which animal and what methods including nucleic acid extraction and PCR method were used. Do the authors also perform IHC? If all samples were negative, how can the authors show that this result is not related to tissue storage / fixation?

Response: The statement regarding the result of retrospective study is confused so we revised the sentence as “A retrospective study of 136 zoo-wild animals derived from 27 different carnivores did not reveal the presence of CPPV-1 genomic antigen as detected by PCR” in page 19, lines 360-361. For the methods and samples used for retrospective study, we have provided the information in the Materials and Methods section as “305 selective fresh samples were included for genomic extraction and identification targeting of the CPPV-1 capsid gene as described above.” in page 11, lines 229-231. Furthermore, we have revised the discussion as the prolonged sample storage may affect the PCR result in as “The negative evidence of retrospective CPPV-1 detection in zoo-wild samples in this study may result from either no close-contact among susceptible animals or, on the other hand, prolonged sample storage may affect the stability of genomic materials..” in page 22, lines 432-434.

Minor comments:

Line 90 "... CPPV-1 reveals FPV is a ...": please change is to as

Line 327 "... and tropism and in kidney...": please delete the second "and".

Response: We have revised them as reviewer suggestions. 

Reviewer #2: Carnivore protoparvovirus-1 (CPPV-1), include feline panleukopenia virus (FPV) and canine parvovirus (CPV), which are widespred among domestic and wild carnivores, causing systemic fatal diseases. Wild fishing cats (Prionailurus viverrinus), is a vulnerable species. Virological (PCR and TEM) and gross and microscopic investigations, identified the presence of an FPV-like parvovirus in fishing cats found dead. Postmortem examination revealed severe enteritis, lymphadenopathy and nephritis. On whole genome sequencing, the virus closely resembled FPV sequences with two peculiar amino acid mutations I566M and M569R in the capsid protein.

The manuscript is of relevant scientific interest. The introduction is well written. The materials and methods seem adequate. The discussion is correct and rather balanced. I have only minor comments.

Response: Thank you for your feeling positive with our manuscripts. 

i) there is a confusion with the terms CPPV-1 and FPV, that are nearly the same thing. I would suggest to use consistently the term FPV-like parvovirus

Response: We have revised as reviewer suggestion. 

ii) there are some parts of the manuscript that could be deleted, reworded or re-phrased.

Abstract: check English. Shorten the final part, very generic.

Line 81-83 it is not clear

Lines 81-88: rephrase

Line 90

Line 316

Line 322: please delete/replace the semicoma (;) after infection

Line 325: FPV is not a variant of CPPV-1

Line 346-347: I would delete the comments (Suggesting… Therefore….)

Line 360-361: rephrase

Line 363: odd sentence (Supporting viral genomic detection…). Rephrase

Response: We have revised as reviewer suggestions. 

iii) A table with the list (and details) of primers used (consensus and specific), even as supplemental file, should be added.

Response: Details of primers used in this study were described in S1_File. 

iv) General comment: the English is good but should be further refined.

Response: The English have been reviewed and revised by native speakers. 

6. PLOS authors have the option to publish the peer review history of their article (what does this mean?). If published, this will include your full peer review and any attached files.

Response: On behalf of corresponding author, I do agree to make public revision. 

Do you want your identity to be public for this peer review? For information about this choice, including consent withdrawal, please see our Privacy Policy.

Reviewer #1: No

Reviewer #2: No

---

## [Decision Letter · Decision Letter 1]

5 Jan 2021

PONE-D-20-33818R1

Natural infection of parvovirus in wild fishing cats (Prionailurus viverrinus) reveals extant viral localization in kidneys

PLOS ONE

Dear Dr. Piewbang,

Thank you for submitting your manuscript to PLOS ONE. After careful consideration, we feel that it has merit but does not fully meet PLOS ONE’s publication criteria as it currently stands. Therefore, we invite you to submit a revised version of the manuscript that addresses the points raised during the review process.

Many thanks for submitting your manuscript to PLOS One

It was reviewed by two experts in the field, and they have recommended some modifications be made prior to acceptance

In particular, please examine the comments regarding the images and the text within the manuscript (which may require inclusion of a pathologist)

I therefore invite you to make these changes and to write a response to reviewers which will expedite revision upon resubmission

I wish you the best of luck with your modifications

Hope you are keeping safe and well in these difficult times

Thanks

Simon

We look forward to receiving your revised manuscript.

Kind regards,

Simon Clegg, PhD

Academic Editor

PLOS ONE

Reviewers' comments:

Reviewer's Responses to Questions

**Comments to the Author**

1. If the authors have adequately addressed your comments raised in a previous round of review and you feel that this manuscript is now acceptable for publication, you may indicate that here to bypass the “Comments to the Author” section, enter your conflict of interest statement in the “Confidential to Editor” section, and submit your "Accept" recommendation.

Reviewer #1: (No Response)

Reviewer #3: (No Response)

2. Is the manuscript technically sound, and do the data support the conclusions?

Reviewer #1: No

Reviewer #3: Partly

3. Has the statistical analysis been performed appropriately and rigorously? 

Reviewer #1: I Don't Know

Reviewer #3: N/A

4. Have the authors made all data underlying the findings in their manuscript fully available?

Reviewer #1: Yes

Reviewer #3: Yes

5. Is the manuscript presented in an intelligible fashion and written in standard English?

Reviewer #1: Yes

Reviewer #3: Yes

6. Review Comments to the Author

Reviewer #1: The manuscript has been improved, however there are still major inconsistencies:

The majority of my comments below deal with the discrepancy between the manuscript text and the displayed photomicrographs and I was wondering, if just these pictures are not representative and the cases show in general, what the authors describe, or if it would be useful to ask a board certified veterinary pathologist reviewing the described findings. This is especially important as the inconsistencies include several organs, e.g. intestine, kidney and spleen. Of particular importance: the picture of the intestine shows a degree of autolysis and I am not convinced that the pictures support the written morphological lesions.

Before publication the authors have to clarify the inconsistency between their morphological description and the pictures included in the manuscript despite the manuscript has been significantly improved compared to the initial submission.

1.) In the abstract the authors state: “Postmortem examination of the carcasses revealed severe inflammation of intestine, lymph node and kidney.”

In their description the authors neither detected inflammation in the intestine nor in the lymph nodes. This description does not fit their morphological description in the results and in the figure legends.

2.) The authors state in the abstract: “CPPV-1 antigen identification in these tissues, using polymerase chain reaction (PCR) and immunohistochemistry (IHC), supported the natural infection of the virus.” How can these techniques support a natural infection?

3.) In the result section the authors state: “Similar degrees of severe lymphoid depletion in spleen were observed in all fishing cats (Fig. 1A).” In the figure description the authors state: “(A) Fishing cat no. 1. Lymphoid depletion of splenic white pulps (asterisks) that indicate by hypocellularity (inset).” These findings are neither visible in the picture nor in the insert. In figure 1A lymphoid follicles are visible and a lymphoid depletion cannot be detected. Furthermore, a severe congestion of the spleen as state in the result section cannot be seen.

4.) With respect to Fig 1B: The findings described by the authors are not visible in the picture: ”Severe diffuse desquamative enteritis, evidenced by shortening villi and necrosis of cryptal epithelium (Fig.1A) was presented in intestinal section of fishing cat no.3 (Fig.1B).”

5.) With respect to Fig 1C: How do the authors know that the immunopositive cells are circulating mononuclear cells or histiocytes as stated in the results/description of figure 1?

6.) The tubular necrosis described by the authors as shown in the pictures is not convincing.

7.) In addition, the authors state: “The PAS staining demostrates the disruption of renal tubular epitheliums while the most architecture of renal tubular basement menbranes were intact (Fig. 2B). This sentence is confusing and the picture especially the insert does not show tubular epithelial cell necrosis. Furthermore, “The tubular lining epithelium are swollen, and the tubular basement membranes are intact (inset).” as described by the authors can not be seen in the pictures.

Reviewer #3: This is an interesting manuscript and one which could be of concern to the fishing cats. I have made a few comments below, but my biggest one is around the pathology images and the text, which do not appear to match the text, so would be grateful if you could check this. But overall it reads well and the comments are only minor, with the above mentioned exception

Line 63-65- there is increasing evidence of CPV in cats (both diseased and asymptomatic), but it is less than in dogs, maybe reword this sentence?

Line 78-79- this reads a little strange- please reword

Line 79- suggested may sound better than warranted

Line 102- comma after cats

Line 112- I think general virological assays isn’t correct- its more molecular or immunological assays

Line 113- comma after cats

Line 118- instructions may be better than suggestions

Line 126- the GAPDH gene of what? Cats or a pathogen/ commensal bacteria?

Line 138- further spelt incorrectly

Line 146-149- this appears unclear and would benefit from some clarification in the text

Line 158-159- repetitive

Line 200- a bootstrap value of 10 000 is often better than 1000

Line 232- it would be nice to know what these species were

Line 238- remove note

Line 248- in the spleen

Figures- I have a few concerns regarding the photos as they do not look representative of what the authors describe. Some show autolysis which makes it difficult to interpret. Do you have any more clear photos. Some of the findings which you mention in the results are not visible in these images

Line 277- in the negative controls

Line 282- you define PAS here despite using it previously, please define at first use

Line 379- co-evolution is suggested for CPV global emergence- what is to say that isn’t the case here rather than pathogen spill over?

7. PLOS authors have the option to publish the peer review history of their article (what does this mean?). If published, this will include your full peer review and any attached files.

Reviewer #1: No

Reviewer #3: No

---

## [Author Response · Author response to Decision Letter 1]

8 Jan 2021

Rebuttal response on PONE-D-20-33818R1

Reviewer #1: The manuscript has been improved, however there are still major inconsistencies:

The majority of my comments below deal with the discrepancy between the manuscript text and the displayed photomicrographs and I was wondering, if just these pictures are not representative and the cases show in general, what the authors describe, or if it would be useful to ask a board-certified veterinary pathologist reviewing the described findings. This is especially important as the inconsistencies include several organs, e.g. intestine, kidney and spleen. Of particular importance: the picture of the intestine shows a degree of autolysis and I am not convinced that the pictures support the written morphological lesions.

Before publication the authors have to clarify the inconsistency between their morphological description and the pictures included in the manuscript despite the manuscript has been significantly improved compared to the initial submission.

Response: Thank you for your kind review. Regarding the morphological descriptions, we kindly requested Dr. Tanit Kasantikul, an American board-certified veterinary pathologist for reviewing and describing the histology of the infected fishing cats. The newly obtained pathological descriptions and some figures were revised according reviewer and the pathologist suggestion. 

1.) In the abstract the authors state: “Postmortem examination of the carcasses revealed severe inflammation of intestine, lymph node and kidney.”

In their description the authors neither detected inflammation in the intestine nor in the lymph nodes. This description does not fit their morphological description in the results and in the figure legends.

Response: We have revised as “Postmortem examination of the carcasses revealed lesions of intestine, spleen and kidney.” Page 2, lines 28-29. 

2.) The authors state in the abstract: “CPPV-1 antigen identification in these tissues, using polymerase chain reaction (PCR) and immunohistochemistry (IHC), supported the natural infection of the virus.” How can these techniques support a natural infection?

Response: We have revised the sentence as “CPPV-1 antigen identification in these tissues, using polymerase chain reaction (PCR) and immunohistochemistry (IHC), supported the infection of the virus.” Page 2, lines 29-30. 

3.) In the result section the authors state: “Similar degrees of severe lymphoid depletion in spleen were observed in all fishing cats (Fig. 1A).” In the figure description the authors state: “(A) Fishing cat no. 1. Lymphoid depletion of splenic white pulps (asterisks) that indicate by hypocellularity (inset).” These findings are neither visible in the picture nor in the insert. In figure 1A lymphoid follicles are visible and a lymphoid depletion cannot be detected. Furthermore, a severe congestion of the spleen as state in the result section cannot be seen.

Response: We revised the Fig. 1A by providing the lower magnification picture to present the spare lymphoid follicles in spleen. Of noted, we have revised the pathological description both in text and figure legend that was addressed by the pathologist as described as “Similar degrees of splenic congestion with few numbers of splenic lymphoid follicle were observed in all fishing cats. The lymphoid follicles were depleted and the remaining lymphoid follicles amid collapsed splenic architecture with increased numbers of prominent splenic trabeculae (Fig 1A). There were scattered karyorrhectic debris of lymphocytes with accumulations of eosinophilic fibrillar materials in the center of such follicle. Few numbers of these lymphocytes contain 5-7 um basophilic intranuclear inclusion bodies that marginate the nuclear chromatin.” in main text (pages 11-12, lines 245-251) and “(A) Fishing cat no. 1. Diffuse congested spleen with sparse numbers of lymphoid follicles. Center of one of the remaining lymphoid follicles contained eosinophilic fibrillar material (fibrin) intermixed with scattered karyorrhectic debris of lymphocytes (lymphocytolysis) (inset). Few numbers of these lymphocytes contained 5-7 um basophilic intranuclear inclusion bodies that marginated the nuclear chromatin (arrow).” in the Fig.1 legend (pages 12, lines 264-268).

4.) With respect to Fig 1B: The findings described by the authors are not visible in the picture: ”Severe diffuse desquamative enteritis, evidenced by shortening villi and necrosis of cryptal epithelium (Fig.1A) was presented in intestinal section of fishing cat no.3 (Fig.1B).”

Response: We have revised the description as “(B) Fishing cat no. 3. Shortening of villi with occasional dilated crypts that contained eosinophilic proteinaceous substances and were lined by markedly attenuated or necrotic crypt epithelial cells (inset). Many crypt epithelial cells were pyknotic and karyorrhectic and rare cells contained similar basophilic intranuclear inclusion bodies (arrows)”. Page 12, lines 268-272.

5.) With respect to Fig 1C: How do the authors know that the immunopositive cells are circulating mononuclear cells or histiocytes as stated in the results/description of figure 1?

Response: We have revised the description as “(C) Fishing cat no. 1. The CPPV-1 immunoreactivity is frequently observed in the cytoplasm of mononuclear cells, where in the area of splenic lymphoid follicle. Page 12, lines 272-274.

6.) The tubular necrosis described by the authors as shown in the pictures is not convincing.

Response: We have changed the Figs. 2A-2B and their legend regarding the pathologist suggestions. 

7.) In addition, the authors state: “The PAS staining demostrates the disruption of renal tubular epitheliums while the most architecture of renal tubular basement membranes were intact (Fig. 2B). This sentence is confusing and the picture especially the insert does not show tubular epithelial cell necrosis. Furthermore, “The tubular lining epithelium are swollen, and the tubular basement membranes are intact (inset).” as described by the authors can not be seen in the pictures.

Response: We have revised the figure and text as “PAS staining demonstrated the renal tubular basement membranes were intact (Fig 2B, inset) (page 13, lines 284-285).

Reviewer #3: This is an interesting manuscript and one which could be of concern to the fishing cats. I have made a few comments below, but my biggest one is around the pathology images and the text, which do not appear to match the text, so would be grateful if you could check this. 

Response: Thank you for you review. We have provided the newly obtained pathological description throughout the pathology pictures, which are examined and reviewed by board-certified veterinary pathologist. 

But overall it reads well and the comments are only minor, with the above mentioned exception

Line 63-65- there is increasing evidence of CPV in cats (both diseased and asymptomatic), but it is less than in dogs, maybe reword this sentence?

Response: We have revised the sentence as “CPV frequently infects animals in the Canidae family and there is increasing evidence of infection in the Felidae family counterpart.” Page 3, lines 265-267.

Line 78-79- this reads a little strange- please reword

Response: We have revised the sentence as “Later, infections of both FPV and CPV variants were reported in leopard cats (9).” Page 4, lines 80-81.

Line 79- suggested may sound better than warranted

Line 102- comma after cats

Response: We have revised as reviewer suggestion. 

Line 112- I think general virological assays isn’t correct- its more molecular or immunological assays

Response: We have revised it as “General virological molecular assays” Page 6, line 115. 

Line 113- comma after cats

Line 118- instructions may be better than suggestions

Response: We have revised as reviewer suggestion. 

Line 126- the GAPDH gene of what? Cats or a pathogen/ commensal bacteria?

Response: We have added the details of GAPDH as “PCR detection specific of the glyceraldehyde-3-phosphate dehydrogenase (GAPDH) gene of feline was used as an internal control as described previously (31). Page 6, lines 128-130.

Line 138- further spelt incorrectly

Response: We corrected it. 

Line 146-149- this appears unclear and would benefit from some clarification in the text

Response: We have revised the text as “Sections of the intestinal tissue of a FPV-infected cat and a CPV-infected dog were used as positive controls, while identical sections incubated with….” Page 7, lines 148-149.

Line 158-159- repetitive

Response: We have corrected the sentence as “The thermal cycling reaction and condition were performed as previously described (33)”. Page 8, lines 160-161. 

Line 200- a bootstrap value of 10 000 is often better than 1000

Response: Thank you for your suggestion, we do agree with your suggestion about higher bootsrapping parameter will be better. However, since genetic diversity among CPPV-1 is relatively low, 1000 repetitive bootstrapping parameter is enough for analysis. 

Line 232- it would be nice to know what these species were

Response: We have provided the list of wildlife carnivores that enrolled in this study in S1 Table. 

Line 238- remove note

Line 248- in the spleen

Response: We have corrected as reviewer suggestion. 

Figures- I have a few concerns regarding the photos as they do not look representative of what the authors describe. Some show autolysis which makes it difficult to interpret. Do you have any more clear photos. Some of the findings which you mention in the results are not visible in these images

Response: Thanks for your concerns, we have revised the figures and their legends following pathologist suggestion. Since the intestinal histological picture seems to be more difficult to interpret due to the fact of tissue autolysis; however, we regret to say that we have only this despite all sections were reviewed by pathologist. 

Line 277- in the negative controls

Line 282- you define PAS here despite using it previously, please define at first use

Response: We have revised as reviewer suggestions. 

Line 379- co-evolution is suggested for CPV global emergence- what is to say that isn’t the case here rather than pathogen spill over?

Response: We have added more details about it in the discussion section as “This result may imply either possible evidence of cross-species transmission between domestic cats and fishing cats or co-evolution of the CPPV-1 variants.” Page 20, lines 397-399.

---

## [Decision Letter · Decision Letter 2]

2 Feb 2021

PONE-D-20-33818R2

Natural infection of parvovirus in wild fishing cats (Prionailurus viverrinus) reveals extant viral localization in kidneys

PLOS ONE

Dear Dr. Piewbang,

Thank you for submitting your manuscript to PLOS ONE. After careful consideration, we feel that it has merit but does not fully meet PLOS ONE’s publication criteria as it currently stands. Therefore, we invite you to submit a revised version of the manuscript that addresses the points raised during the review process.

Many thanks for submitting your manuscript to PLOS One

It was reviewed by two experts in the field, and they have recommended some further minor modifications be made prior to acceptance

I therefore invite you to make these changes and to write a response to reviewers which will expedite revision upon resubmission

I wish you the best of luck with your modifications

Hope you are keeping safe and well in these difficult times

Thanks

Simon

We look forward to receiving your revised manuscript.

Kind regards,

Simon Clegg, PhD

Academic Editor

PLOS ONE

Reviewers' comments:

Reviewer's Responses to Questions

**Comments to the Author**

1. If the authors have adequately addressed your comments raised in a previous round of review and you feel that this manuscript is now acceptable for publication, you may indicate that here to bypass the “Comments to the Author” section, enter your conflict of interest statement in the “Confidential to Editor” section, and submit your "Accept" recommendation.

Reviewer #1: (No Response)

Reviewer #3: All comments have been addressed

2. Is the manuscript technically sound, and do the data support the conclusions?

Reviewer #1: Yes

Reviewer #3: Yes

3. Has the statistical analysis been performed appropriately and rigorously? 

Reviewer #1: N/A

Reviewer #3: Yes

4. Have the authors made all data underlying the findings in their manuscript fully available?

Reviewer #1: Yes

Reviewer #3: Yes

5. Is the manuscript presented in an intelligible fashion and written in standard English?

Reviewer #1: Yes

Reviewer #3: Yes

6. Review Comments to the Author

Reviewer #1: The authors did a great job and significantly improved the manuscript. However, there are still some issues:

In the materials and methods (Animals and routine postmortem examination section) the authors state: "Selective vital organs including lung, liver, heart, spleen, and kidney were sampled from all fishing cats, while the intestine and mesenteric lymph node were additionally collected from fishing cat no.3." while in the (full-length genetic characterization of fishing cat CPPV-1) section the authors write: "Briefly, the extracted nucleic acids obtained from intestine, spleen and kidneys of two fishing cats were individually amplified using a GoTaq® Hot Start Green Master Mix (Promega, Madison, WI, U.S.A.) and specific primers (S1 File)."

 How can you isolate nucleic acids from the intestine of two cats when samples are only available from cat 3? Please clarify.

In the description of figure 1 the size of the scale bars is missing.

In the abstract in line 29 I suggest to change ... revealed lesions of intestine ... to revealed lesions in intestine ...

Reviewer #3: The manuscript is much improved from a previous version, and the pathology images are much better. A few minor issues are detailed below, but I fully expect that this can be accepted when these minor issues are addressed

Line 26- are widely spread (add in word)

Line 27- a globally vulnerable …. (reword)

Line 30- infection by the virus (reword)

Line 67- this needs a reference, and some good references here for cats infected with CPV- both symptomatic and asymptomatic

Line 81- replace of, with as the

Within your materials and methods you take some samples from some animals, but not from others, yet the samples not taken appear later in the study for the PCR analysis. Can you please check this just for clarity?

Line 408- determinant may sound better than determination?

7. PLOS authors have the option to publish the peer review history of their article (what does this mean?). If published, this will include your full peer review and any attached files.

Reviewer #1: No

Reviewer #3: No

---

## [Author Response · Author response to Decision Letter 2]

2 Feb 2021

Rebuttal response on PONE-D-20-33818R2

Reviewer #1: The authors did a great job and significantly improved the manuscript. However, there are still some issues:

In the materials and methods (Animals and routine postmortem examination section) the authors state: "Selective vital organs including lung, liver, heart, spleen, and kidney were sampled from all fishing cats, while the intestine and mesenteric lymph node were additionally collected from fishing cat no.3." while in the (full-length genetic characterization of fishing cat CPPV-1) section the authors write: "Briefly, the extracted nucleic acids obtained from intestine, spleen and kidneys of two fishing cats were individually amplified using a GoTaq® Hot Start Green Master Mix (Promega, Madison, WI, U.S.A.) and specific primers (S1 File)."

 How can you isolate nucleic acids from the intestine of two cats when samples are only available from cat 3? Please clarify.

Response: Thank you for your positive review on our manuscript. There is a missing information regarding the samples used for genome sequencing and we do revise them as “the extracted nucleic acids obtained from spleen and kidneys of two fishing cats plus additional intestinal tissue of fishing cat no. 3, were…” in page 9, lines 183-184. 

In the description of figure 1 the size of the scale bars is missing.

Response: the size of the scale bars in Fig. 1 is already described as found in page 13, line 276. 

In the abstract in line 29 I suggest to change ... revealed lesions of intestine ... to revealed lesions in intestine ...

Response: We revised it as reviewer suggestion. 

Reviewer #3: The manuscript is much improved from a previous version, and the pathology images are much better. A few minor issues are detailed below, but I fully expect that this can be accepted when these minor issues are addressed

Line 26- are widely spread (add in word)

Line 27- a globally vulnerable …. (reword)

Line 30- infection by the virus (reword)

Response: Thank you for your recommendation, we have revised them as reviewer suggestions. 

Line 67- this needs a reference, and some good references here for cats infected with CPV- both symptomatic and asymptomatic

Response: We have provided the potential references as your suggestion as found in page 3, line 67.

Line 81- replace of, with as the

Within your materials and methods you take some samples from some animals, but not from others, yet the samples not taken appear later in the study for the PCR analysis. Can you please check this just for clarity?

Response: There is some missing information regarding the samples used for genome sequencing of the CPPV-1, so we have revised the sentence as “the extracted nucleic acids obtained from spleen and kidneys of two fishing cats plus additional intestinal tissue of fishing cat no. 3, were…” in page 9, lines 183-184. 

Line 408- determinant may sound better than determination?

Response: We revised it as reviewer suggestion.

---

## [Editor Report · Decision Letter 3]

4 Feb 2021

Natural infection of parvovirus in wild fishing cats (Prionailurus viverrinus) reveals extant viral localization in kidneys

PONE-D-20-33818R3

Dear Dr. Piewbang,

We’re pleased to inform you that your manuscript has been judged scientifically suitable for publication and will be formally accepted for publication once it meets all outstanding technical requirements.

Kind regards,

Simon Clegg, PhD

Academic Editor

PLOS ONE

Additional Editor Comments:

Many thanks for resubmitting your manuscript to PLOS One

As you have addressed all the comments and the manuscript reads well, I have recommended it for publication

You should hear from the Editorial Office shortly.

It was a pleasure working with you and I wish you the best of luck for your future research

Hope you are keeping safe and well in these difficult times

Thanks

Simon

---

## [Editor Report · Acceptance letter]

22 Feb 2021

PONE-D-20-33818R3 

Natural infection of parvovirus in wild fishing cats (*Prionailurus viverrinus*) reveals extant viral localization in kidneys 

Dear Dr. Piewbang:

I'm pleased to inform you that your manuscript has been deemed suitable for publication in PLOS ONE. Congratulations! Your manuscript is now with our production department. 

Kind regards, 

on behalf of

Dr. Simon Clegg 

Academic Editor

PLOS ONE